# Screw sense excess and reversals of helical polymers in solution

Francisco Rey-Tarrío [1,2], Rafael Rodríguez [1,2], Emilio Quiñoá [1,2] & Félix Freire [1,2] ✉

The helix reversal is a structural motif found in helical polymers in the solid state, but whose existence is elusive in solution. Herein, we have shown how the photochemical electrocyclization (PEC) of poly(phenylacetylene)s (PPAs) can be used to determine not only the presence of helix reversals in polymer solution, but also to estimate the screw sense excess. To perform these studies, we used a library of well folded PPAs and different copolymers series made by enantiomeric comonomers that show chiral conflict effect. The results obtained indicate that the PEC of a PPA will depend on the helical scaffold adopted by the PPA backbone and on its folding degree. Then, from these studies it is possible to determine the screw sense excess of a PPA, highly important in applications such as chiral stationary phases in HPLC or asymmetric synthesis.

Dynamic helical polymers have attracted the attention of the scientific community during the last decades due to their stimuli-responsive properties and the functionality attributed to the helical scaffold. Thus, while helical sense and elongation of dynamic helical polymers can be altered by their interaction with external stimuli[1–21], the helical scaffolds adopted by the polymers are responsible of their applications in different fields such as asymmetric synthesis[22–24], chiral recognition[25], or chiral stationary phases in HPLC[26–28]. Importantly, to create a dynamic helical polymer with a good performance in certain applications it is necessary to know its secondary structure, which also allows establishing a structure/function relationship in this kind of materials. Among dynamic helical polymers, poly(phenylacetylene)s (PPAs) has been extensively studied due to the robustness of the polymerization reaction with Rh(I) catalysts, which can be done in polar and non-polar solvents, with phenylacetylene monomers bearing different polar and non-polar functional groups[29–31]. In all cases, polymers with high stereoregularity, *cis* configuration of double bonds, and in high yield are obtained. The large library of PPAs found in literature allows establishing that depending on the size and position of the pendant group in the phenylacetylene monomer it is possible to create PPAs

with different elongations or $\omega_1$ (dihedral angle between conjugated double bonds)[32,33]. To determine an approximate value for $\omega_1$, the combination of the information obtained from different structural techniques −UV-vis, circular dichroism (CD)[34–36], vibrational circular dichroism (VCD)[37], Raman[38], Raman optical activity (ROA)[39], differential scanning calorimetry (DSC)[40] and atomic force microscopy (AFM)−[41–49] is needed. This fact is due to the complexity of the helical scaffolds found in PPAs, which are made by two coaxial helices: an internal helix described by the polyene backbone and an external one described by the pendants, which, depending on the stereoregularity of the polyene backbone, can rotate in the same (cis-cisoidal, $\omega_1 < 90°$) or opposite directions (cis-transoidal, $\omega_1 > 90°$)[50–52]. Recently, our group reported another approach to get information about the secondary structure of PPAs[53]. In such a case, photochemical electrocyclization (PEC) of the polyene backbone is used[40,54], where the half-time $t_{1/2}$ of the process to reach a null ECD signal in the vinylic region in solution depends directly on the $\omega_1$ adopted by the PPA (Fig. 1). Thus, a dilute solution of a PPA [c.a. $1·10^{-3}$ M] is irradiated with visible light ($\lambda > 350$ nm) and the CD signal decay monitored versus time to afford the PEC rate constant and the $t_{1/2}$ as the relevant parameters using a first order reaction fitting. Next, by

[1]Centro Singular de Investigación en Química Biolóxica e Materiais Moleculares (CiQUS), University of Santiago de Compostela, 15782 Santiago de Compostela, Spain. [2]Departamento de Química Orgánica, University of Santiago de Compostela, 15782 Santiago de Compostela, Spain. ✉e-mail: felix.freire@usc.es

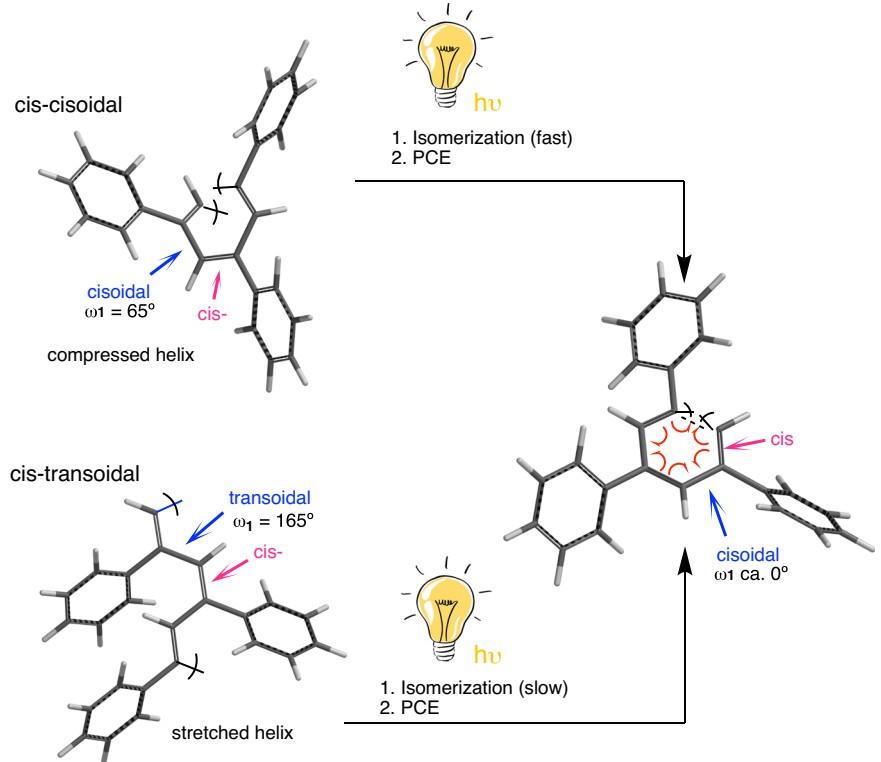

**Fig. 1 | Photoisomerization process.** Graphical illustration of the photoisomerization process needed in cis-cisoidal and cis-transoidal PPAs prior the photochemical electrocyclization.

applying Eq. (1) is possible to obtain an approximate value for $\omega_1$.

$$\omega_1 = -147.5 + 312.6 * (1 - e^{\wedge}(-4.659 \cdot 10^{-2} * t_{1/2})) \tag{1}$$

Interestingly, although these studies allow to model an approximate secondary structure of the PPA, no information about the presence of reversals within a helical scaffold can be extracted. Reversals are structural motifs found in helical polymers that induce helical inversions along polymer chains. In PPAs, Prof. Yashima visualized this structural motif by AFM studies of a PPA bearing α-aminoisobutyric acid (Aib) n-decyl esters as achiral pendant[55]. This motif was also visualized by AFM in chiral PPAs possessing a screw sense excess[56]. Curiously, although the existence of helix reversals was evidenced in the solid state through high-resolution AFM images, the study of this structural motif is solution has not yet been addressed. This fact is due to the complexity of the polymers and the absence of robust structural tools that allows determining the secondary structure of polymers in solution.

In this work, we address the identification of helix reversals in solution using the PEC of PPAs. As it was mentioned above, the rate of the PEC of PPAs depends on the elongation ($\omega_1$) of the polymer and on its dynamic behaviour. Therefore, the presence of mismatches such as helix reversals within a helical scaffold must have effects on the PEC of PPAs. To perform these studies, we choose a combination of polymers and copolymers with different folding propensities induced by solvents or by chiral-to-chiral communication produced between comonomers and, using PEC as the tool of choice, we elucidate the secondary structure of PPAs and their screw sense excess in solution.

## Results
### Selection of polymers and their characteristics
As homopolymers with known secondary structure, we have chosen PPAs with well-known cis-cisoidal (c-c) and cis-transoidal (c-t) scaffolds

and with different benzamide and anilide pendant groups. For instance, poly-(R)-**1** bears the *para*-ethynylbenzamide of (R)-phenylglycine methyl ester as pendant. This polymer adopts a c-t scaffold −$\omega_1$ c.a. 148°, helical pitch 3.8 nm− with a $P_{int}/M_{ext}$ screw sense excess in low-polar solvents such as THF and CHCl₃ (Fig. 2a, b)[57,58]. Although the folding degree is different based on the intensity of the ECD trace, which is attributed to the presence of reversals along the helix, that could alter the rate of the PEC process. Therefore, two vials containing 7 mL of poly-(R)-**1** (c = 1.02·10⁻³ M) in THF and CHCl₃ were irradiated with visible light, and the PEC process was monitored by plotting the ECD signal decay versus time (Fig. 2c). The half-life ($t_{1/2}$ = ln2/k) was obtained for poly-(R)-**1**(CHCl₃) $t_{1/2}$ = 63 min, while for poly-(R)-**1**(THF) the data did not fit. In that case, we could estimate the value of $ECD_{50\%}$ from the graph ($t_{ECD50\%}$= 10 min). These data indicate a c-t scaffold in CHCl₃ ($\omega_1$ c.a. 148°) when fitted to Eq. 1, while in THF the fast isomerization process does not point to a c-t scaffold. As expected, the $\omega_1$ obtained from irradiation studies of poly-(R)-**1** in CHCl₃ (well folded), is in good agreement with the value obtained experimentally from other structural techniques. While the value obtained for this polymer in THF (poorly folded) is far from the admissible, based in the expected $t_{1/2}$ rates for a c-t scaffold in the photochemical electrocyclization process. This result indicates that there is a strong relationship between the photochemical electrocyclization process and the folding of the helical polymer.

Therefore, two structural parameters are closely related in the PEC studies, the helical scaffold, and the folding degree. Thus, a misfolded PPA will produce small half-life values of the PEC process leading to an incorrect $\omega_1$ value. Therefore, to avoid wrong $\omega_1$ assignment based on the folding degree of the PPA, it is necessary to determine first if the polymer adopts a c-c or a c-t scaffold. From previous studies, it is known that PEC highly depends on the helical scaffold adopted by the PPA, being fast for c-c and slow for c-t PPAs. We believe that this effect is produced by the different orientations of conjugated double bonds in both scaffolds, that are well oriented in c-c

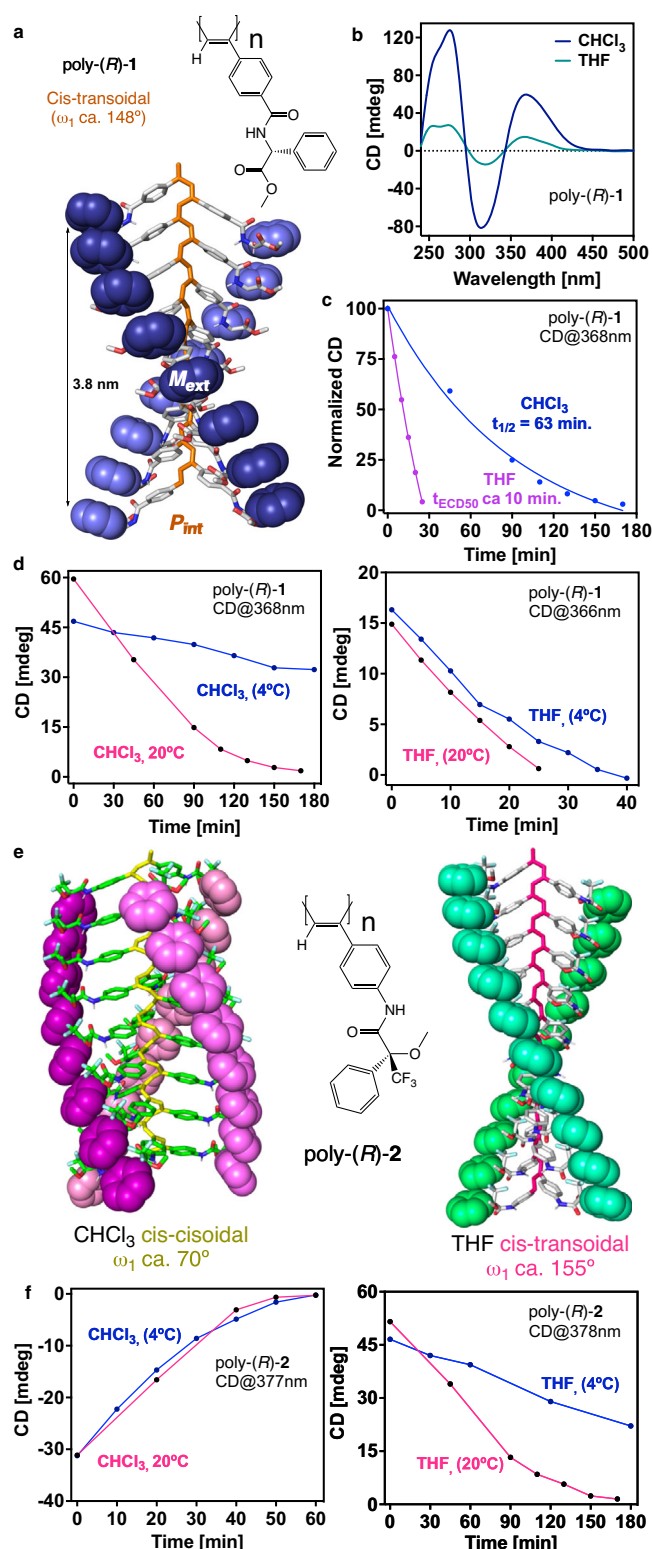

**Fig. 2 | Structures of poly-(R)-1 and of poly-(R)-2 and ECD signal decay vs. time.**
**a** Chemical and 3D structure of poly-(R)-1[58]. **b** ECD spectra of poly-(R)-1 in CHCl₃ and THF (blue and green lines, respectively). **c** Normalized CD signal decay vs. time for poly-(R)-1 at different irradiation times under visible light in CHCl₃ and THF (pink and blue lines, respectively). **d** ECD signal decay vs. time for poly-(R)-1, in CHCl₃ and THF, at different irradiation times under visible light at two temperatures (20 °C pink line, 4 °C blue line). **e** Chemical and 3D models of poly-(R)-2[58]. **f** ECD signal decay vs. time for poly-(R)-2, in CHCl₃ and THF, at different irradiation times under visible light at two temperatures (20 °C pink line, 4 °C blue line). [poly-(R)-1]= 1.02·10⁻³ M, [poly-(R)-2]= 9.00·10⁻⁴ M.

to promote the PE, while in c-t PPAs, the polyene backbone needs to reorient the conjugated double bonds to promote the PE (Fig. 1). These conformational changes should be affected by temperature, being the PEC delayed at low temperatures.

## Photochemical electrocyclization studies

To corroborate this hypothesis, PEC studies at room temperature (rt) and 4 °C were carried out for poly-(R)-1 and for poly-(R)-2, that bears the *para*-ethynylanilide of the (R)-α-methoxy-α-tri-fluoromethylphenylacetic acid as pendant. This polymer adopts a compressed helix in CHCl₃ (3.0 nm helical pitch, $\omega_1$ c.a. 70°), while a stretched helix is induced in poly-(R)-2 when dissolved in THF (3.9 nm helical pitch, $\omega_1$ c.a. 155°; Fig. 2e)[50,58].

Moreover, PEC studies indicate a good folding for poly-(R)-2 in both solvents providing an $\omega_1$ value that matches the one obtained from the combination of other structural techniques −poly-(R)-2 $\omega_{1(THF)}$ c.a. 155°, $\omega_{1(CHCl3)}$ c.a. 72°−.

Thus, vials containing 7 mL of poly-(R)-1 (c = 1.02·10⁻³ M) and poly-(R)-2 (c = 9.00·10⁻⁴ M) prepared in THF and CHCl₃, respectively, were irradiated with visible light at rt and 4 °C. These studies show how temperature affects c-c and c-t scaffolds differently. Thus, while the PEC of poly-(R)-2 dissolved in CHCl₃ is not affected by temperature changes −c-c scaffold−, the PEC of poly-(R)-2 in THF and the PEC of poly-(R)-1 in CHCl₃ and THF are delayed at low temperatures −c-t helix − (Fig. 2d, f).

To extrapolate this observation to all PPAs, we performed PEC studies at rt and 4 °C for other PPAs with well-known c-c and c-t secondary structures. These studies show that in the case of PPAs possessing a c-c helical scaffold such as poly-(S)−3 and poly-(R)−4 that bear the *para*-ethynylanilide of (S)-mandelic acid as pendant and the *para*-ethynylanilide of (S)-α-methoxy-α-phenylacetic acid as substituent respectively, the ECD signal decay produced during the studies reach an ECD = 0 at the same time for both temperatures, rt and 4 °C (Fig. 3a, b). On the other hand, in c-t PPAs −poly-(S)-5, poly-(S)-6 and poly-(S)-7, bearing the *para*-ethynylbenzamide of (L)-alanine methyl ester, (L)-valine methyl ester and (L)-phenylalanine methyl ester respectively−, the ECD signal decay produced by PEC of the polyene backbone is largely affected by temperature reaching an ECD = 0 to longer times (Fig. 3a, c). Therefore, from these studies we can state that when the half-time obtained from PEC at rt and 4 °C is the same or almost the same at both temperatures, the PPA adopts a cis-cisoidal structure. On the other hand, if the half-time increases notably at lower temperatures, the PPA adopts a cis-transoidal scaffold.

Next, we aimed to explore the role of the reversals in the PEC studies of random copolymer series containing the two enantiomers of monomer-1 −poly[(R)-1ᵣ-co-(S)-1₍₁₋ᵣ₎]− (Fig. 4a, b). ECD studies show a chiral conflict between the two enantiomeric comonomers, which induces opposite helical senses within the copolymer chain (Fig. 4c, d). Therefore, on this copolymer series, a controlled number of reversals (screw sense excess) is introduced along the chain when a mismatch of absolute configuration −R/S− is present. Thus, in a poly[(R)-1₀.₉-co-(S)-1₀.₁]), 90% of the comonomers induce a P helix −[(R)-1]−, while 10% of the comonomers induce an M helix −[(S)-1]−. As a result, a screw sense excess of 80% towards the P helix is present in the copolymer. Therefore, the % screw sense excess of poly[(R)-1₀.₈-co-(S)-1₀.₂], poly[(R)-1₀.₇-co-(S)-1₀.₃], poly[(R)-1₀.₆-co-(S)-1₀.₄] and poly[(R)-1₀.₅-co-(S)-1₀.₅] are 60%, 40%, 20% and 0% respectively (Fig. 4e, f).

PEC studies of poly[(R)-1ᵣ-co-(S)-1₍₁₋ᵣ₎] copolymer series in CHCl₃ − c = 1.02·10⁻³ M, 7 mL− show a strong dependence between the copolymer photostability and the screw sense excess (% sse) −$t_{ECD50\%}$ or $t_{ECD20\%}$ vs (100 -% sse)− where the decay can be fitted to a first order reaction (2).

$$Y = (Y_0 - \text{Plateau}) \cdot e^{(-K \cdot X)} + \text{Plateau} \qquad (2)$$

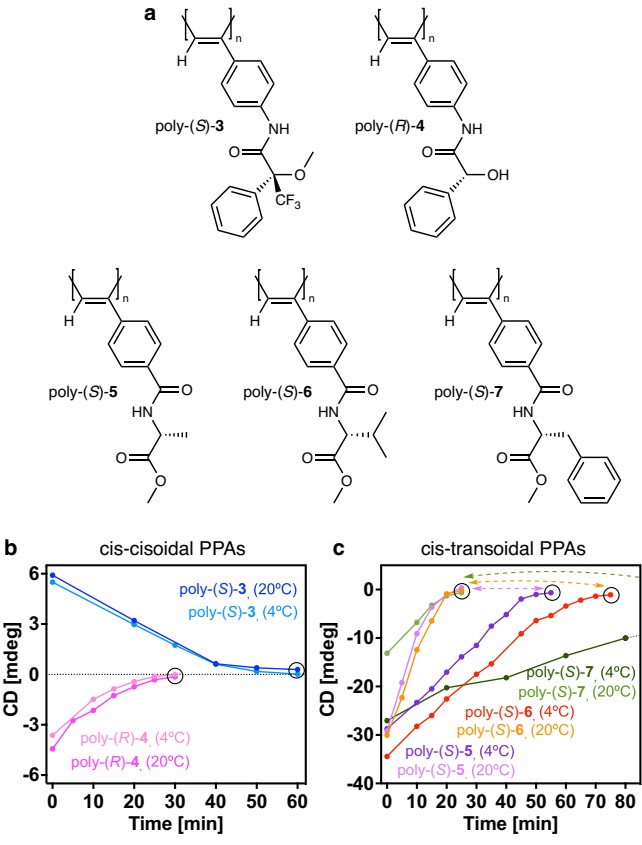

**a**

poly-(*S*)-**3**

poly-(*R*)-**4**

poly-(*S*)-**5**

poly-(*S*)-**6**

poly-(*S*)-**7**

**b** cis-cisoidal PPAs

poly-(*S*)-**3** (20°C)
poly-(*S*)-**3** (4°C)
poly-(*R*)-**4** (4°C)
poly-(*R*)-**4** (20°C)

**c** cis-transoidal PPAs

poly-(*S*)-**7** (4°C)
poly-(*S*)-**7** (20°C)
poly-(*S*)-**6** (4°C)
poly-(*S*)-**6** (20°C)
poly-(*S*)-**5** (4°C)
poly-(*S*)-**5** (20°C)

**Fig. 3 | Structures of cis-cisoidal and cis-transoidal PPAs and ECD signal decay vs. time. a** Chemical structure of poly-(*S*)-**3**, poly-(*R*)-**4**, poly-(*S*)-**5**, poly-(*S*)-**6** and poly-(*S*)-**7**. **b** ECD signal decay vs. time for poly-(*S*)-**3** (THF, CD@386 nm at 20 and 4 °C, pink and light pink lines, respectively) and poly-(*R*)-**4** (THF, CD@386 nm at 20 and 4 °C, blue and light blue lines, respectively) at different irradiation times under visible light at two temperatures. **c** Idem for poly-(*S*)-**5** (CHCl₃, CD@368 nm at 20 and 4 °C, light purple and purple lines, respectively), poly-(*S*)-**6** (CHCl₃, CD@367 nm at 20 and 4 °C, orange and red lines, respectively) and poly-(*S*)-**7** (CHCl₃, CD@368 nm at 20 and 4 °C, light green and green lines, respectively). [poly-(*S*)-**3**]= 1.19·10⁻³ M, [poly-(*R*)-**4**]= 1.13·10⁻³ M, [poly-(*S*)-**5**]= 1.30·10⁻³ M, [poly-(*S*)-**6**]= 1.16·10⁻³ M, [poly-(*S*)-**7**]= 9.76·10⁻⁴ M.

Where Y is the time needed to reduce the ECD signal to the 50% or 20%. $Y_0$ is the time needed to reach a CD = 50% or 20% in the homopolymer (100% sse), Plateau is the minimum time need to unfold the copolymer if it has a 0 % sse (ECD = 0), K is the rate constant, and X is the axially racemic fraction found in the PPA [100 · screw sense excess (% sse)].

$$(c - t_{145-150})t_{ECD50\%} = 59.84 \cdot e^{(-0.001906 \cdot (100 - \%sse))} - 4.530 (R^2 = 0.979) \quad (3)$$

$$(c - t_{145-150})t_{ECD20\%} = 86.82 \cdot e^{(-0.02776 \cdot (100 - \%sse))} + 7.857 (R^2 = 0.985) \quad (4)$$

Using this equation, it is possible to extract an approximate value for the screw sense excess of poly-(*R*)-**1** when dissolved in THF. In this case, the $t_{ECD50\%}$ and $t_{ECD20\%}$ values obtained during the PEC studies are 11 min and 19 min, respectively, which corresponds to the presence of ca. 35% and 40% of screw sense excess within the polymer chain.

To further confirm the robustness of this protocol that allows determining an approximated value of the screw sense excess within a PPA, a random copolymer series containing the two enantiomers of monomer-**2** was prepared −poly[(*R*)-**2**ᵣ-co-(*S*)-**2**₍₁₋ᵣ₎]−. This polymer is well folded in solution as happens with poly-(*R*)-**1**. In case of poly-(*R*)-**2**,

two different scaffolds are obtained when the polymer is dissolved in chloroform and THF, where the polymer adopts a c-c ($\omega_1$ c.a. 65-75°) and a c-t ($\omega_1$ c.a. 155°) helical structure respectively (Fig. 5a, b).

ECD studies show a chiral conflict between the two enantiomeric comonomers within the copolymer series in both solvents, CHCl₃ and THF. The lack of communication between the two enantiomeric monomers indicates the presence of a controlled number of reversals in the poly[(*R*)-**2**ᵣ-co-(*S*)-**2**₍₁₋ᵣ₎] copolymer series, which can be tuned by varying the (*R*)-**2**/(*S*)-**2** ratio (Fig. 5c, d).

PEC studies were carried out for the different copolymers of the poly[(*R*)-**2**ᵣ-co-(*S*)-**2**₍₁₋ᵣ₎] copolymer series and monitored by ECD (Fig. 6a, c). Similarly to the results obtained for the poly[(*R*)-**1**ᵣ-co-(*S*)-**1**₍₁₋ᵣ₎] copolymer series, a strong relationship is found between the copolymer photostability and the screw sense excess −$t_{ECD50\%}$ or $t_{ECD20\%}$ vs 100 · %sse− within the poly[(*R*)-**2**ᵣ-co-(*S*)-**2**₍₁₋ᵣ₎] copolymer series (Fig. 6b, d), where the decay can be fitted to a first-order reaction (2). Two different equations are obtained depending on the helical scaffold adopted by poly−**2**, a cis-transoidal helix in THF, and a more compressed cis-cisoidal structure in CHCl₃. This equation is different from those obtained for the poly[(*R*)-**1**ᵣ-co-(*S*)-**1**₍₁₋ᵣ₎] copolymer series (3,4), because the elongation degree is different ($\omega_1$ c.a. 145-150° for poly-**1** and $\omega_1$ c.a. 70° for poly−**2** in CHCl₃ and c.a. 155-160° for poly−**2** in THF).

$$(c - t_{155})t_{ECD50\%} = 132.3 \cdot e^{(-0.004616 \cdot (100 - \%sse))} - 69.53 (R^2 = 0.966) \quad (5)$$

$$(c - t_{155})t_{ECD20\%} = 76.71 \cdot e^{(-0.01533 \cdot (100 - \%sse))} + 21.24 (R^2 = 0.986) \quad (6)$$

$$(c - c)t_{ECD50\%} = 15.62 \cdot e^{(-0.02262 \cdot (100 - \%sse))} + 5.512 (R^2 = 0.989) \quad (7)$$

$$(c - c)t_{ECD20\%} = 18.73 \cdot e^{(-0.03734 \cdot (100 - \%sse))} + 16.05 (R^2 = 0.984) \quad (8)$$

This fact indicates, as predicted, that the PEC depends on the dynamic behaviour of the polymer and on the secondary structure of the PPA.

### Screw sense excess in PPAs with c-c or c-t scaffolds

Considering the library of PPAs with well-known secondary structure that we handle, we have postulated three different equations that will allow us predicting screw sense excess in PPAs with c-c ($\omega_1$ ca. 60-80°, Eq. 3) or c-t helix scaffolds ($\omega_1$ ca. 140-150° Eq. 4, $\omega_1$ ca. 150-155° Eq. 5).

With this information on hand, we decided to analyse the %sse present in PPAs with known secondary structure that fit those values.

Thus, poly-(*S*)-**3** and poly-(*R*)-**4** bearing the *para*-ethynylanilides of (*S*)-mandelic and (*S*)-α-methoxy-α-phenylacetic acids, respectively as substituents were chosen as examples of PPAs with a c-c scaffold to perform these studies. PEC studies of these polymers monitored by ECD allowed us to extract their polymer photostability values ($t_{ECD50\%}$) −poly-(*S*)-**3**, $t_{ECD50\%}$= 21 min; poly-(*R*)-**4**, $t_{ECD50\%}$= 8.8 min− (Fig. 6e), which are correlated with the percentage of reversals present in the helix scaffold by fitting them to Eq. 5. As a result, it was concluded that poly-(*S*)-**3** is fully folded in THF (%sse<1 %), while poly-(*R*)-**4** shows approximately 69% along the helical scaffold when dissolved in THF. Similarly, 20% of remaining ECD signal −poly-(*S*)-**3**, $t_{ECD20\%}$= 34 min; poly-(*R*)-**4** $t_{ECD20\%}$= 17 min−is correlated with the %sse present in the helix scaffold by fitting the data to Eq. 5. As a result, poly-(*S*)-**3** is fully folded in THF (%sse= 1%), while poly-(*R*)-**4** shows approximately 80% along the helix scaffold when dissolved in THF.

Analogous studies were carried out for poly-(*S*)-**5**, poly-(*S*)-**6** and poly-(*S*)-**7**, three PPAs that bear the *para*-ethynylbenzamide of (*L*)-alanine methyl ester, (*L*)-valine methyl ester and (*L*)-phenylalanine methyl ester respectively (Fig. 6f). These PPAs possess a c-t helix scaffold with

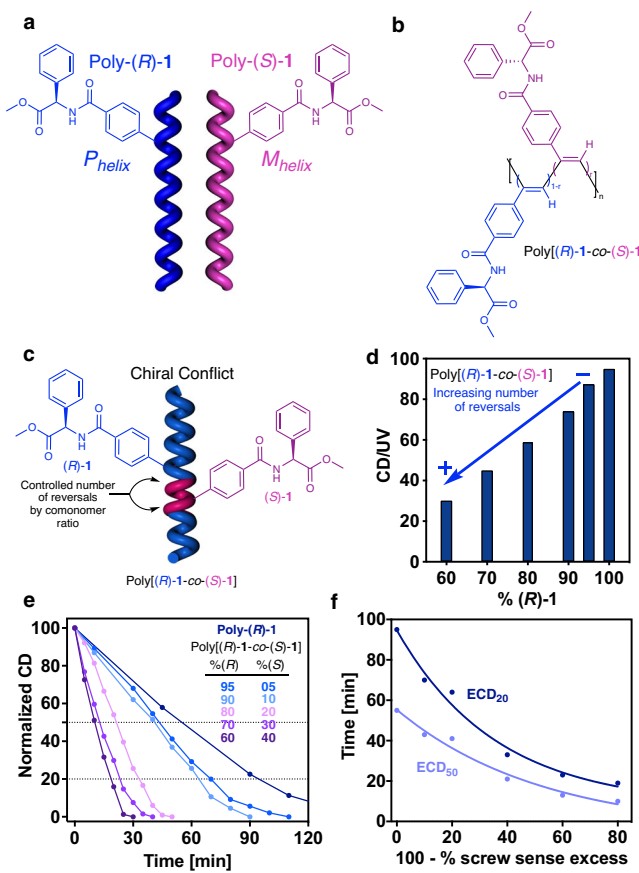

**Fig. 4 | Chiral conflict phenomenon. a** Chemical structure and helix sense representation of poly-(R)-**1** and poly-(S)-**1**. **b** Chemical structure of the copolymer poly[(R)-**1**-co-(S)-**1**]. **c** Conceptual representation of the chiral conflict phenomenon. **d** Variation of the CD/UV signal (CD@368 nm) vs percentage of (R)-**1** in the poly[(R)-**1**-co-(S)-**1**] series. **e** Normalized CD signal decay (CD@368 nm) vs. time for poly[(R)-**1**-co-(S)-**1**] series at different irradiation times under visible light (dark blue to purple gradient color palette). **f** Time consumed for reach ECD signal 50% and 20% during the photochemical electrocyclization in the poly[(R)-**1**-co-(S)-**1**] series (light blue and dark blue lines, respectively). [poly[(R)-**1**-co-(S)-**1**]]= $1.02 \cdot 10^{-3}$ M.

$\omega_1$ of ca. 148°. To determine the %sse in these polymers it is mandatory to apply Eq. (3). For these polymers PEC studies show a fast isomerization of the polyene backbone that does not correspond to a fully folded helical scaffold. For poly-(S)-**5**, $t_{ECD50\%}$= 7.3 min, while for poly-(S)-**6**, $t_{ECD50\%}$= 8.7 min and for poly-(S)-**7**, $t_{ECD50\%}$= 10 min that correspond to helices with a 85%, 79% and 74% of screw sense excess within the helix respectively. Similarly, at 20% of remaining ECD signal, poly-(S)-**5**, $t_{ECD20\%}$= 13 min, while for poly-(S)-**6**, $t_{ECD20\%}$= 15 min and poly-(S)-**7**, $t_{ECD20\%}$= 15 min that corresponds to helices with a 100%, 90% and 90% of screw sense excess within the helix respectively. Furthermore, the good agreement between percentages obtained applying $t_{ECD50\%}$ or $t_{ECD20\%}$ evidence the robustness of the results.

Importantly, although these polymers are ECD active and show an intense ECD trace, the photochemical studies reveal a poorly folded structure that will badly affect to the potential applications of PPAs such as chiral ligands in asymmetric synthesis or chiral stationary phases in HPLC among others.

Finally, to cover the entire range of c-t PPAs we need prepare a c-t copolymer with ($\omega_1 > 160°$; for $\omega_1$ ca. 140-150° we propose Eq. 4 and for $\omega_1$ ca. 150-155° the Eq. 5). Poly-(R)-**8** that bears the *meta*-ethynylbenzamide of (R)-phenylglycine methyl ester as pendant is constituted by an equilibrium of two cis-transoidal helices with different elongation (both $P_{int}$, $\omega_1$ ca. 160° (CD@373 nm) and 165° (CD@428

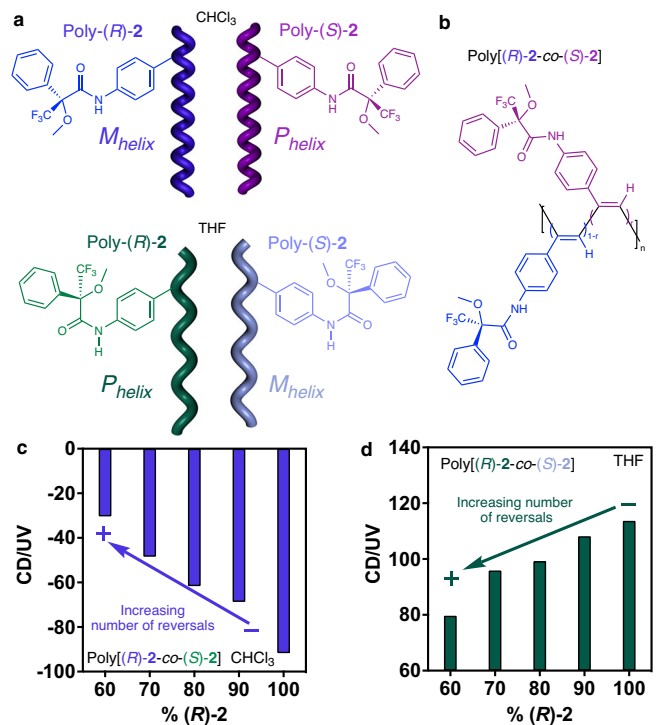

**Fig. 5 | Studies with random copolymer series. a** Chemical structure and helix sense/elongation representation of poly-(R)-**2** and poly-(S)-**2** in CHCl₃ and THF. **b** Chemical structure of the copolymer poly[(R)-**2**-co-(S)-**2**]. **c, d** Variation of the CD/UV signal vs. percentage of (R)-**2** in the poly[(R)-**2**-co-(S)-**2**] series in CHCl₃ (CD@377 nm) and THF (CD@378 nm), respectively (blue and green bars, respectively).

nm). To explore the role of reversals on stretched c-t polymers, copolymer series containing the two enantiomers of monomer-**8** − poly[(R)-**8**ᵣ-co-(S)-**8**₍₁₋ᵣ₎]− was polymerized (Fig. 7a, b). ECD studies for the 165° helix (CD@428 nm) show a chiral conflict between the two enantiomeric comonomers (Fig. 7c). Therefore, a controlled number of reversals is introduced again along the chain based on the mismatch of absolute configuration −R/S− present.

PEC studies, monitored by ECD, for the different copolymers of poly[(R)-**8**ᵣ-co-(S)-**8**₍₁₋ᵣ₎] reveals a strong relationship between the copolymer photostability and the percentage of reversals (Fig. 7d). Such as poly[(R)-**1**ᵣ-co-(S)-**1**₍₁₋ᵣ₎] and poly[(R)−**2**ᵣ-co-(S)−**2**₍₁₋ᵣ₎] series, the decay can be fitted to a first order reaction (9) (Fig. 7e). Due to the presence of a mixtures of two helices, only the time at 20% of ECD signal was considered. Furthermore, for the dependence of the PEC on the dynamic behaviour, the effect of the reversals in the photostability is reduced as we increase $\omega_1$ −elongation and dynamic behaviour go oppositely in PPAs−.[33] Thus, representation $t_r/t_i$ at ECD = 20% demonstrate that the effect of the reversals in the photostability decrease as $\omega_1$ increase (Fig. 7f). Being $t_r$ the ECD signal at 20% for the copolymer with a 100-%sse and $t_i$ the ECD signal at 20% for the corresponding homopolymer.

$$(c - t_{165})t_{ECD20\%} = 40.8 \cdot e^{(-0.02261 \cdot (100 - \%sse))} + 178.5 (R^2 = 0.983) \quad (9)$$

## Discussion

We have demonstrated that the PEC of PPAs in the solution can be used as a structural tool to determine the secondary structure of PPAs and their screw sense excess. PEC of PPAs does not depends on the chemical structure of the pendant groups or the solvents used to do the studies, being the rate of the PEC process affected only by the secondary structure of the PPA. Thus, while in c-c PPAs the PEC is fast, in

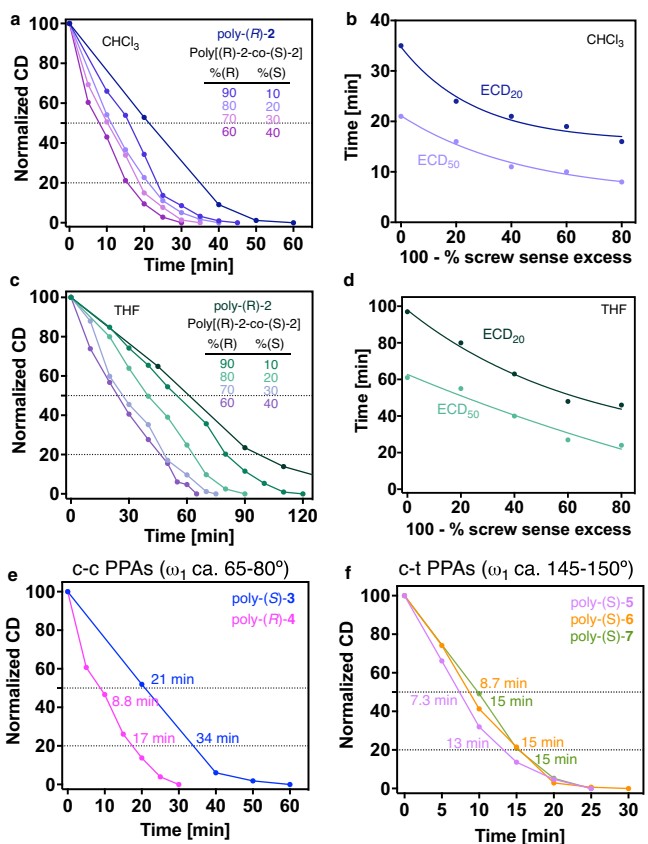

**Fig. 6 | Studies at different irradiation times under visible light. a, c** Normalized CD signal decay vs. time for poly[(*R*)-**1**·co-(*S*)-**1**] series at different irradiation times under visible light in CHCl₃ (CD@377 nm, dark blue to purple gradient color palette) and THF (CD@378 nm, dark green to purple gradient color palette) respectively. **b, d** Time consumed for reach ECD signal 50 % and 20 % during the photochemical electrocyclization vs. % of reversals in the poly[(*R*)-**1**·co-(*S*)-**1**] series in CHCl₃ (CD@377 nm; light and dark blue, respectively) and THF (CD@378 nm; light and dark green, respectively) respectively. [poly[(*R*)-**2**·co-(*S*)-**2**]]= 9.00·10⁻⁴ M. Normalized CD signal decay vs. time at different irradiation times under visible light for (**e**) poly-(*S*)-**3** (CD@386 nm, blue line) and poly-(*R*)-**4** (CD@383 nm, pink line) and (**f**) poly-(*S*)-**5** (CD@368 nm, pink line), poly-(*S*)-**6** (CD@367 nm, orange line) and poly-(*S*)-**7** (CD@368 nm, green line). [poly-(*S*)-**3**]= 1.19·10⁻³ M, [poly-(*R*)-**4**]= 1.13·10⁻³ M, [poly-(*S*)-**5**]= 1.30·10⁻³ M, [poly-(*S*)-**6**]= 1.16·10⁻³ M, [poly-(*S*)-**7**]= 9.76·10⁻⁴ M.

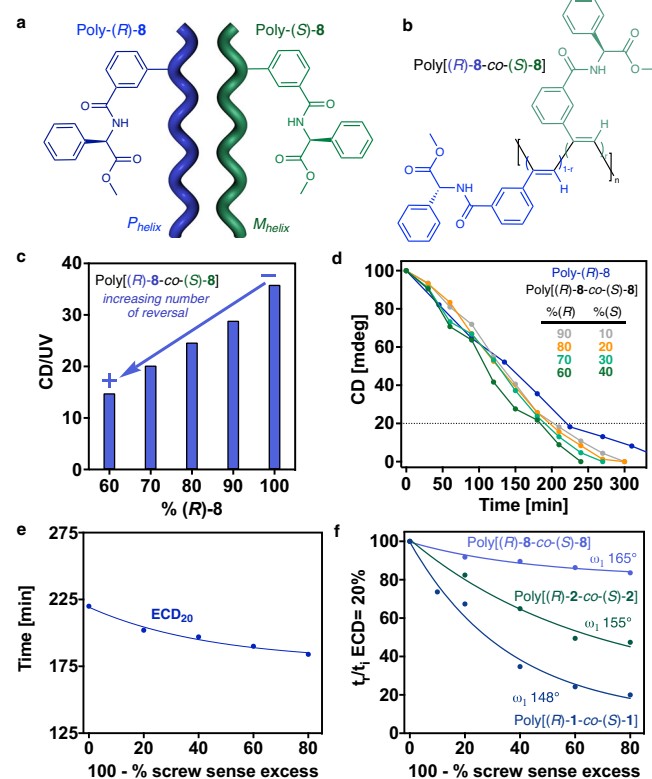

**Fig. 7 | Role of reversals on stretched c-t polymers. a** Chemical structure and helix sense representation of poly-(*R*)-**8** and poly-(*S*)-**8**. **b** Chemical structure of the copolymer poly[(*R*)-**8**·co-(*S*)-**8**]. **c** Variation of the CD/UV signal (CD@428 nm) vs percentage of (*R*)−**8** in the poly[(*R*)-**8**·co-(*S*)−**8**] series. **d** Normalized CD signal decay (CD@428 nm) vs. time for poly[(*R*)-**8**·co-(*S*)-**8**] series at different irradiation times under visible light (dark blue to green gradient color palette). **e** Time consumed for reach ECD signal 20% during the photochemical electrocyclization vs (100 -%sse) in the poly[(*R*)-**8**·co-(*S*)-**8**] series. **f** Normalized time at ECD = 20% vs (100 -%sse) for [poly[(*R*)-**1**·co-(*S*)-**1**] (dark blue line), [poly[(*R*)-**2**·co-(*S*)-**2**] (dark green line) and [poly[(*R*)-**8**·co-(*S*)-**8**]] (light purple line). [poly[(*R*)−**1**·co-(*S*)−**1**]]= 1.02·10⁻³ M, [poly[(*R*)-**2**·co-(*S*)-**2**]]= 9.00·10⁻⁴ M, [poly[(*R*)-**8**·co-(*S*)−**8**]]= 1.02·10⁻³ M.

c-t PPAs the PEC is slower, speed being the speed largely affected by temperature. Therefore, PEC at different temperatures allow us to classify PPAs into c-c or c-t PPAs. Moreover, PEC studies also allow obtaining the dihedral angle between conjugated double bonds in well-folded polymers. However, if the polymers are not well folded, the presence of reversals along the helix affects largely the PEC process, being this shortened in time. This relationship between the PEC of PPAs and screw sense excess along the helix allowed us to generate an equation that predicts the percentage of screw sense excess present in a PPA with a certain scaffold. To apply this equation, it is necessary to know if the polymer has c-c or c-t structure, which can be easily obtained from PEC studies at different temperatures or by using DSC studies. In the case of c-c PPAs there is only one equation to determine the folding of PPAs, and in the case of c-t PPAs is necessary to have an approximated value for ω₁ to determine its folding degree. At this stage, there are three different equations for c-t PPAs; one for PPAs with ω₁ ca. 148°; a second one for PPAs with ω₁ ca. 155°, and a third one for PPAs with ω₁ ca. 165°.

These studies can be of great interest to the scientific community because they are dealing with the folding degree of PPAs in solution. In literature, dynamic helical polymers are used as chiral catalyst, chiral recognition agents and chiral stationary phases in HPLC. However, these polymers are applied without considering the polymer folding. By using this approximation, it is possible to determine if the polymer is well folded or not in solution, and therefore, to explain how the helical scaffold affects the applications of the polymer.

## Methods

### General procedure for polymerization

The reaction flask (sealed ampoule) was dried under vacuum and argon flushed for three times before monomer was added as a solid. Then, the flask was evacuated on a vacuum line and flushed with dry argon (three times). Dry THF was added with a syringe and the triethylamine dropwise. A solution of rhodium norbornadiene chloride dimer, [Rh(nbd)Cl]₂, in THF was added at 30 °C. The reaction mixture was stirring at 30 °C for 6 h. Then, the resulting polymer was diluted in CH₂Cl₂ and it was precipitated in a large amount of methanol, centrifuged (2 times), reprecipitated in hexane and centrifuged again. As exception, in the centrifugation of poly-(*S*)-**3**, it is used a mixture MeOH:Ether (10:90) instead pure methanol.

### General protocol for PPA irradiation

To perform the photochemical electrocyclization studies, we employed the methodology depicted in Scheme S1 (see below). 0.3 mg/mL solutions of the corresponding polymers/copolymers were

prepared, potential acid traces present in $CHCl_3$ were removed by filtering the solvent through basic alumina.

The solutions were deoxygenated prior to light irradiation by bubbling Ar for 5 min. 0.2 mL of purged solution of the corresponding polymer were transferred to a 1 mm cuvette to measure CD/UV-Vis spectra −being this spectrum t = 0 min of irradiation− and 7.0 mL of solution was transferred to a second flask under Ar atmosphere. Irradiation experiments were performed using a collimator Lens (x1.0) of a MAX-303 (Asahi Spectra) equipped with a UV-VIS mirror module (300-600 nm) using a short pass filter (VIS 550 nm 25 dia.). After the 20 min required for the irradiation source stabilization, the light beam (with the intensity selected at 100%) was directly focused to the flask containing the 7.0 mL of purged polymer solution. The irradiation system was covered to avoid the incidence of any additional light. Following the abovementioned procedure, the polymer solutions were irradiated for the desired times, recording after each time the CD spectrum by transferring aliquots of 200 μL of polymer solution to a 1 mm cuvette, via syringe, keeping the system under an inert atmosphere and without interrupting the irradiation process. For polymers measured on 10 mm cuvettes, the irradiation process is halted by closing the irradiation path of the light source. Next, 2.0 mL of the corresponding polymer solution was taken via syringe and CD/UV-Vis spectra were consequently measured. After measurement, the polymer solution is reintroduced into the flask where the irradiation process takes place, and the path light is open again. Note that the time while the light source remains closed is not taken into consideration in the total time. Remarkably, without light irradiation the reaction stops immediately, and the obtained CD trace remains constant.

## Data availability

The data that support the findings of this study is available within the manuscript, its supplementary information file, or from the corresponding author upon request.

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

## Acknowledgements

Financial support from MINECO (PID2019-109733GB-I00, E. Q. and F. F.) and Juan de la Cierva Incorporación contract (IJC2020-042689-I, R. R.), Xunta de Galicia (ED431C 2022/21, Centro Singular de Investigación de Galicia acreditación 2019-2022, ED431G 2019/03, E. Q. and F. F.; and a predoctoral fellowship for F. R. T) and the European Regional Development Fund (ERDF) is gratefully acknowledged.

## Author contributions

F.F. conceived the project. F.R.T. and R.R. synthesized, characterized monomers and polymers, and performed PEC studies. F.F. and E.Q.

secured funding and supervised the overall project. The manuscript was prepared through the contribution of all the authors.

## Competing interests

The authors declare no competing interests.
