## [Peer Review File · Nature Communications]

Screw Sense Excess and Reversals of Helical Polymers in SolutionREVIEWER COMMENTS

Reviewer #1 (Remarks to the Author):

Freire and coworkers present their investigation on the reversibility of helical PPAs, and observed that photochemical electrocyclization (PE) of PPAs in solution was controlled by the helical scaffold and folding. Generally speaking, this work provides an effective tool to determine the helical folding of PPAs and greatly advances the understanding of the relationship between chirality and molecular structures. Based on these findings, I think this work is worthwhile for publication in NC. However, the formulation of the structural-related chirality reversibility was much based on too many assumptions, therefore, authors are required to pay attention to the following comments for revision:

- a. In the equation (2), the percentage of reversals (X) of the homopolymer is defined as 0%, and the percentage of reversals of the copolymer is directly derived from the feeding ratio. Meanwhile, the parameter "Plateau" is set to "the minimum time need to unfold the copolymer if it has a 100% of reversals". Based on above set of assumptions, are the results of these calculations acceptable? In addition, the original data to calculate equation 3-6 should be given in the supplementary information.
- b. In order to apply the proper equation, it is necessary to estimate the dihedral angle between conjugated double bonds (ω_1) of polymer backbone in solution. How can we determine the ω_1 of PPAs in solution due to the possible reversals that cause equation (1) to be invalid?
- c. Poly-(R)-2 adopts a c-c or c-t helical scaffold in CHCl_3 or THF, respectively. However, the change in contracted and stretched helices did not result in a drastic change in the absorbance corresponding to the chromophores from polyene backbone as in other PPAs [the helix oscillation of poly(arylacetylene)s by Tabata, the poly(3,5-disubstituted phenylacetylene)s by Aoki and Wan]. Furthermore, how does the conjugation degree of the backbone and the angle of the benzene ring to the backbone contribute to the absorption peak of the polyene backbone?
- d. It is necessary to provide the molecular weight of the polymers or provide appropriate discussion in the SI, even though molecular weight has no apparent effect on the PE as demonstrated in the authors' previous work (Angew. Chem. Int. Ed. 2021).
- e. There are some errors in the manuscript. for example,
 1. The half-life given in Figure 1c is clearly unreasonable.
 2. Wrong molecular structure and ambiguous NMR spectrum in Figure S4.

Reviewer #2 (Remarks to the Author):

The paper by Freire and coworkers describes how to determine the rate and presence of helical reversals in classes of poly(phenyl acetylenes). This work primarily relies on circular dichroism studies to establish the rate of photo-electrocyclization of the polymer backbone, which the authors hypothesize correlates well with the folding of the polymer in solution. The authors investigate this hypothesis with a series of PPAs, and are able to demonstrate that the rate of PE correlates well with the secondary structure of the polymer and the % of reversals in the polymer backbone. Given the utility of these studies, and the fact that this is the first time that helix reversal of dynamic helical polymers can be measured in solution, I think that this work fits well in Nature Communications and should be accepted after major revisions.

The authors mention that the half time for poly-(R)-1(CHCl_3) is 63 min but omit poly-(R)-1 (THF). From Figure 1c, the $t_{1/2}$ is labeled as 30 min for THF but the value should be around 10-15 min based on the plot. This brings up a question, why "a poorly folded PPA will produce small half-life values of the PE process"? Where does this conclusion come from? This is the basis of the whole research but seems to be lacking evidence.

How did the authors calculate the concentrations and what are the molecular weights of these polymers? Can the monomer enantiomers be copolymerized with the same rate/chance/selectivity? For example, is it possible that the two enantiomers have less tendency to incorporate the other one during polymerization so the polymer formed is actually a mixture of poly-(R) and poly-(S) rather than poly-(R)-co-(S)?

A figure in the introduction would help to conceptualize/visualize the ω value that is important to characterizing the secondary structure, and another figure would be useful to demonstrate the photoelectrocyclization reactions mentioned. I can hypothesize the reactions that may occur, but it would be very useful to a reader to show these reactions in a figure given the importance to the paper.

PE is a common abbreviation for polyethylene – given this is a polymer-centric work, PE should be re-abbreviated to another abbreviation (e.g., PEC) that cannot be confused with polyethylene.

As a matter of style, it is more accessible to use the formalism e^x instead of $\exp(x)$

Editing for typos is needed. For example, “stablishing” in the introduction should be “establishing”.

It is unclear how the 3D models of the polymer in Figure 1A and 1E were obtained – are these simulations? If so, a reference should be listed to the paper in which these simulations are reported, or short of that, a computational details section should be created in the SI describing the simulations that were performed to obtain these (exquisite) 3D structures.

In the SI, the authors do not list a source for their $[\text{Rh}(\text{nbd})\text{Cl}]_2$. A source should be listed either as a reference to a paper or as a mention of a chemical supplier in the materials and methods section.

Reviewer 1:

Reviewer 1 states: - In the equation (2), the percentage of reversals (X) of the homopolymer is defined as 0%, and the percentage of reversals of the copolymer is directly derived from the feeding ratio. Meanwhile, the parameter “Plateau” is set to “the minimum time need to unfold the copolymer if it has a 100% of reversals”. Based on above set of assumptions, are the results of these calculations acceptable? In addition, the original data to calculate equation 3-6 should be given in the supplementary information.

Our answer: As the reviewer mentions, the data was not expressed correctly. We have now denoted the percentage of reversals as screw sense excess, which is a more appropriate term. Thus, if the homopolymer is fully folded (100% P or M) the screw sense excess is 100%. Therefore, in the plateau, that is when the polymer is axially racemic (50% P and 50% M), the screw sense excess is 0%.

Therefore, in the copolymer series, the X axis denoted as % of reversals was changed to % of screw sense excess which is calculated as follows: In a poly[(R)- $\mathbf{1}_{0.9}$ -co-(S)- $\mathbf{1}_{0.1}$] 90% of the comonomers induce a P helix —[(R)- $\mathbf{1}$]—, while 10% of the comonomers induces a M helix —[(S)- $\mathbf{1}$]—, as a result a screw sense excess of 80% towards the P helix is present in the copolymer. Therefore, the % screw sense excess of the copolymers poly[(R)- $\mathbf{1}_{0.8}$ -co-(S)- $\mathbf{1}_{0.2}$], poly[(R)- $\mathbf{1}_{0.7}$ -co-(S)- $\mathbf{1}_{0.3}$], poly[(R)- $\mathbf{1}_{0.6}$ -co-(S)- $\mathbf{1}_{0.4}$] and poly[(R)- $\mathbf{1}_{0.5}$ -co-(S)- $\mathbf{1}_{0.5}$] is 60%, 40%, 20% and 0% respectively. A new paragraph explaining how to determine the screw sense excess in a copolymer was introduced in the main text (page 4).

The original data to calculate equations 3-6 was added to the SI.

Reviewer 1 states: - In order to apply the proper equation, it is necessary to estimate the dihedral angle between conjugated double bonds (ω_1) of polymer backbone in solution. How can we determine the ω_1 of PPAs in solution due to the possible reversals that cause equation (1) to be invalid?

Our answer: To determine the screw sense excess of a PPA, it is necessary to know an approximate value for ω_1 . To do this, you can use a combination of structural techniques such as AFM, ECD or DSC among others, or as we demonstrate in this work, by performing PCE studies at different temperatures e.g., 4 and 20°C. If the half-time obtained in both cases is the same or almost the same, the PPA adopts a *cis-cisoidal* structure. In that case, only one equation is applied to determine the folding degree. On the other hand, if the half-time increases notably at lower temperatures, the PPA adopts a *cis-transoidal* scaffold. In such case, we must know an approximated value of ω_1 —ca. 150, 160, 165—, because the isomerization rate of conjugated double bonds is largely affected by this value in this kind of scaffolds. This value must be extracted from other structural techniques.

A new paragraph has been included in the revised manuscript (page 4): *Therefore, from these studies we can state that when the half-time obtained from PEC at rt and 4°C is the same or almost the same at both temperatures, the PPA adopts a cis-cisoidal structure. On the other*

hand, if the half-time increases notably at lower temperatures, the PPA adopts a cis-transoidal scaffold.

Reviewer 1 states: - Poly-(R)-2 adopts a c-c or c-t helical scaffold in CHCl₃ or THF, respectively. However, the change in contracted and stretched helices did not result in a drastic change in the absorbance corresponding to the chromophores from polyene backbone as in other PPAs [the helix oscillation of poly(arylacetylene)s by Tabata, the poly(3,5-disubstituted phenylacetylene)s by Aoki and Wan]. Furthermore, how does the conjugation degree of the backbone and the angle of the benzene ring to the backbone contribute to the absorption peak of the polyene backbone?

Our answer: As the reviewer mentions, the absorbance of the polyene backbone in cis-cisoidal and cis-transoidal PPAs with a ω_1 dihedral angle smaller than 150° is very similar. This fact is due to the poor conjugation of the p-orbitals of conjugated double bonds. When this dihedral angle is higher than 150°, the conjugation of p-orbitals increases and large bathochromic shifts are observed, which are higher as ω_1 becomes larger. This is the reason why poly-(R)-2 have similar UV absorbance in both solvents although the value of ω_1 is very different. For a detailed structural characterization of poly-(R)-2 see *Chem. Sci.* 4, 2735-2743 (2013).

On the other hand, we have demonstrated that the dihedral angle of the benzene ring to the backbone can contribute to the absorption, where depending on its value an extra Cotton band can appear in the ECD spectrum [please see reference 39: *Angew. Chem. Int. Ed.* 59, 9080-9087 (2020)]. However, this dihedral angle does not affect the photoisomerization process of conjugated double bonds during the PCE. The rate of this process is affected by the dynamic behavior of the PPA and the dihedral angle between conjugated double bonds.

Reviewer 1 states: - It is necessary to provide the molecular weight of the polymers or provide appropriate discussion in the SI, even though molecular weight has no apparent effect on the PE as demonstrated in the authors' previous work (*Angew. Chem. Int. Ed.* 2021).

Our answer: The molecular weight of the homopolymers and copolymers used in these studies were measured by GPC and added to the SI (see page S29).

Reviewer 1 states: -There are some errors in the manuscript. for example,
1. The half-life given in Figure 1c is clearly unreasonable.
2. Wrong molecular structure and ambiguous NMR spectrum in Figure S4.

Our answer: Half-life of Figure 1c and the molecular structure and NMR spectrum of Figure S4 have been corrected.

Reviewer 2:

Reviewer 2 states: The authors mention that the half time for poly-(R)-1(CHCl₃) is 63 min but omit poly-(R)-1 (THF). From Figure 1c, the $t_{1/2}$ is labeled as 30 min for THF but the value should be around 10-15 min based on the plot. This brings up a question, why “a poorly folded

PPA will produce small half-life values of the PE process”? Where does this conclusion come from? This is the basis of the whole research but seems to be lacking evidence.

Our answer: Figure 1c has been corrected, the $t_{\text{ECD}50\%}$ is 10 min as the reviewer mentions.

The basis of the research was addressed in the introduction, when we mention “Herein, we want to address the identification of helix reversals in solution using the PEC of PPAs. As it was mentioned above, the rate of the **PEC of PPAs depends on the elongation (ω_1) of the polymer and on its dynamic behavior**. Therefore, the presence of **mismatches such as helix reversals within a helical scaffold should have effects on the PEC** of PPAs.”

From previous studies we know how the secondary structure affects the PEC of PPAs. Herein, we want to analyze how the presence of mismatches within the helix scaffold affects the isomerization rate of PPAs during the PEC.

***Reviewer 2 states:** How did the authors calculate the concentrations and what are the molecular weights of these polymers? Can the monomer enantiomers be copolymerized with the same rate/chance/selectivity? For example, is it possible that the two enantiomers have less tendency to incorporate the other one during polymerization so the polymer formed is actually a mixture of poly-(R) and poly-(S) rather than poly-(R)-co-(S)?*

Our answer: The copolymer concentration is calculated considering the average molecular weight of the two components present in the copolymer and their ratio. Molecular weight of copolymers was measured by GPC, and the values were added to the SI (see GPC studies page S29).

PEC studies were carried out in a mixture of homopolymers poly-(R)-**1** and poly-(S)-**1** in a 7/3 ratio. The results obtained are different than those obtained for the copolymer, indicating that the copolymers are random copolymers. In the mixtures of homopolymers, during the first minutes the ECD spectra remains unaltered due to the identical isomerization of both homopolymers with opposite ECD signature. Once the less abundant polymer (poly-(S)-**1**) is consumed, an ECD trace decay is observed due to the PEC process suffered by the remaining poly-(R)-**1**, which is ECD active. A new section was added to the SI with these studies, see “Irradiation studies on a mixture of homopolymers” page S28.

***Reviewer 2 states:** A figure in the introduction would help to conceptualize/visualize the ω value that is important to characterizing the secondary structure, and another figure would be useful to demonstrate the photoelectrocyclization reactions mentioned. I can hypothesize the reactions that may occur, but it would be very useful to a reader to show these reactions in a figure given the importance to the paper.*

Our answer: A figure highlighting the relationship between the secondary structure of the PPA and the photoelectrocyclization process has been added as scheme 1. For a detailed mechanism of the PEC, please see reference 40.

***Reviewer 2 states:** PE is a common abbreviation for polyethylene – given this is a polymer-centric work, PE should be re-abbreviated to another abbreviation (e.g., PEC) that cannot be*

confused with *polyethylene*.
As a matter of style, it is more accessible to use the formalism e^x instead of $\exp(x)$.

Our answer: We agree with the reviewer and the photochemical electrocyclization abbreviation was changed to PEC. In addition, we also changed in the equations $\exp(x)$ by e^x .

Reviewer 2 states: *Editing for typos is needed. For example, “stablishing” in the introduction should be “establishing”.*

Our answer: Typos have been corrected.

Reviewer 2 states: *It is unclear how the 3D models of the polymer in Figure 1A and 1E were obtained – are these simulations? If so, a reference should be listed to the paper in which these simulations are reported, or short of that, a computational details section should be created in the SI describing the simulations that were performed to obtain these (exquisite) 3D structures.*

Our answer: Structural studies to elucidate the secondary structure of poly-(*R*)-**1** and poly-(*R*)-**2** are found in references 57 and 50 respectively. From these studies, we obtained the values of the main dihedral angles, such as ω_1 . Then, an oligomer of poly-(*R*)-**1** and poly-(*R*)-**2** were built using Spartan'20 (SPARTAN, Wavefunction, Inc., 18401, Von Karman Avenue, Suite 370, Irvine CA 92612 USA) and Pymol (*PyMOL*, Schrödinger, L., & DeLano, W. (2020)]. The software used to build and visualize the 3D model of poly-(*R*)-**1** and poly-(*R*)-**2** have been introduced as reference 58 in the manuscript.

Reviewer 2 states: *In the SI, the authors do not list a source for their $[\text{Rh}(\text{nbd})\text{Cl}]_2$. A source should be listed either as a reference to a paper or as a mention of a chemical supplier in the materials and methods section.*

Our answer: The chemical supplier for the $[\text{Rh}(\text{nbd})\text{Cl}]_2$ catalyst is Sigma Aldrich (Merck KGaA). The name of the supplier was added to the Si in the materials and methods section.

REVIEWERS' COMMENTS

Reviewer #1 (Remarks to the Author):

The authors have addressed my concerns properly and revised the manuscript accordingly. Therefore, I feel the manuscript in its present state is suitable for publication.

Reviewer #2 (Remarks to the Author):

The manuscript has improved significantly. All comments of the reviewers have been addressed. There are still some minor typos/edits needed (for example ET3N should be N_ET₃ in Tables 1 to 4 of the supplementals or dispersity is not anymore written as PDI but the D with the bar through it) but I have no problem recommending acceptance.